# New land-use change scenarios for Brazil: Refining global SSPs with a regional spatially-explicit allocation model

**Francisco Gilney Silva Bezerra**[1]*, **Celso Von Randow**[1], **Talita Oliveira Assis**[1], **Karine Rocha Aguiar Bezerra**[1], **Graciela Tejada**[1], **Aline Anderson Castro**[1], **Diego Melo de Paula Gomes**[1], **Rodrigo Avancini**[1], **Ana Paula Aguiar**[1,2]

**1** General Cordination of Earth Sciences, National Institute for Space Research (INPE), São José dos Campos, SP, Brazil, **2** Stockholm Resilience Centre, Stockholm University, Stockholm, Sweden

☯ These authors contributed equally to this work.
* franciscogilney@gmail.com

**Data Availability Statement:** All regional scenarios files are available from the "LuccME/INLAND land-use scenarios for Brazil 2050" database (https://zenodo.org/record/5123560).

## Abstract

The future of land use and cover change in Brazil, particularly due to deforestation and forest restoration processes, is critical for the future of global climate and biodiversity, given the richness of its five biomes. These changes in Brazil depend on the interlink between global factors due to its role as one of the main exporters of commodities globally and the national to local institutional, socioeconomic, and biophysical contexts. Aiming to develop scenarios that consider the balance between global (e.g., GDP growth, population growth, per capita consumption of agricultural products, international trade policies, and climatic conditions) and local factors (e.g., land use, agrarian structure, agricultural suitability, protected areas, distance to roads, and other infrastructure projects), a new set of land-use change scenarios for Brazil were developed that aligned with the global structure Shared Socioeconomic Pathways (SSPs) and Representative Concentration Pathway (RCPs) developed by the global change research community. The narratives of the new scenarios align with SSP1/RCP 1.9 (Sustainable development scenario), SSP2/RCP 4.5 (Middle of the road scenario), and SSP3/RCP 7.0 (Strong inequality scenario). The scenarios were developed by combining the LuccME spatially explicit land change allocation modeling framework and the INLAND surface model to incorporate the climatic variables in water deficit. Based on detailed biophysical, socioeconomic, and institutional factors for each biome in Brazil, we have created spatially explicit scenarios until 2050, considering the following classes: forest vegetation, grassland vegetation, planted pasture, agriculture, a mosaic of small land uses, and forestry. The results aim to detail global models regionally. They could be used regionally to support decision-making and enrich the global analysis.

## 1 Introduction

Land available for agricultural expansion is an increasingly scarce resource in several global regions [1, 2]. The expansion of the agricultural frontier, which is currently concentrated in the tropics [3, 4], affects the regulation of the hydrological and climatic regime, local

**Funding:** This study was supported by awards from the Amazon Fund (14209291) and the São Paulo Research Foundation – FAPESP (2017/22269-2). The National Council for Scientific and Technological Development – CNPq also awarded a grant to CVR (314780/2020-3).

**Competing interests:** NO authors have competing interests.

**Abbreviations:** The following abbreviations are used in this manuscript: AIC, Akaike Information Criterion; CLUE, Conversion of Land Use and its Effects; GDP, Gross Domestic Product; HIC, High-Income Countries; IAM, Integrated Assessment Models; IBGE, Brazilian Institute of Geography and Statistics; IMAGE, Integrated Model to Assess the Global Environment; INLAND, Integrated Surface Process Model; IPCC, Intergovernmental Panel on Climate Change; IL, Indigenous Lands; LIC, Low-Income Countries; LR, Legal Reserves; LUCC, Land Use and Land Cover Change; LuccME, Land Use and Land Cover Change Modeling Framework; LuccMEBR, Land Use and Land Cover Change Model to Brazil; MIC, Middle-Income Countries; PA, Protected Areas; PPA, Permanent Protection Areas; RCP, Representative Concentration Pathway; SPA, Shared climate Policy Assumptions; SSP, Shared Socioeconomic Pathways; UC, Conservation Units of Integral Protection and Sustainable Use.

socioeconomic relations and generates great biodiversity loss. This context could be aggravated if we consider population increase and projected demand for food in 2050 (25% and 40%, respectively) [5].

In this context, global models and scenarios, particularly those quantified with Integrated Assessment Models (IAMs), which represent complex interactions and feedback on a long-term scale between the socioeconomic system (including climate policies) and the natural system [6], which play a key role in helping us to understand the impacts and consequences of agricultural expansion in different regions. In Brazil, for example, this process over the last few decades has contributed to the country consolidating worldwide as one of the main commodity-exporting countries, whether agricultural or mineral. One of the key impacts of this process is the loss of natural vegetation in the Amazon and Cerrado biomes. On the other hand, other areas in Brazil, such as the Atlantic Forest, are undergoing a forest transition [7]. Integrating and understanding the factors that influence land use and land cover change (LUCC) in Brazil in different regions are important for defining indicators for guiding public policies to establish sustainable development strategies.

Global models and scenarios may fail to capture the regional dynamics of land change. They do not always include local factors, regional narratives, the national political and institutional framework, and the dynamics and magnitude of intraregional factors that determine the demand for land. In addition, the information used in most global models is aggregated for comparability across large regions, such as continents [8–10]. In this sense, Dala-Nora [8] points out that a balance between global and local factors is necessary, as the integration of these complex factors that act on a global and regional scale through extensive flow networks can change the structure and consistency of land-use change scenarios. Furthermore, van Vuuren [11] pointed out that studies that examine phenomena of a more precise scale should consider more detailed information (e.g., geographic characteristics, land use patterns, or the location of cities).

In this paper, we present a new set of land change scenarios for Brazil, aligned with the global Shared Socioeconomic Pathways (SSPs) and Representative Concentration Pathway (RCPs) framework developed by the global change research community [9, 12–16] to support the Intergovernmental Panel on Climate Change (IPCC). These scenarios are being widely used and downscaled to several regions (e.g., [16–20]), and adopting them as a reference allows us to link our scenarios to the global context better. The regionalized scenarios developed here aim to represent the diversity of processes linked to land-use change in the Brazilian territory. The modeling approach considers Brazilian biomes' interregional socio-ecological differences, including a more detailed analysis scale, without losing global relations. We modeled changes in natural vegetation, large and small-scale agricultural lands, and planted forests. These land change processes are directly related to regional and local factors. The global context still plays a significant role in these processes.

## 2 Materials and methods

### 2.1 Overall conceptualization and structure

For the development of regionalized scenarios, three levels or spatial scales were considered: (i) global: comprises information from around the world and seeks to integrate into regional scenarios the contribution of factors such as GDP growth, population growth, per capita consumption of agricultural products, international trade policies, and climatic conditions; (ii) regional or national: corresponds to the Brazilian territory and seeks to integrate intraregional drivers such as national demand, institutions, and governance, economic and technological development, etc.; and (iii) local, which aggregates spatial drivers to the local, regional scale,

such as land use, agrarian structure, agricultural suitability, protected areas, distance to roads, and other infrastructure projects. The integration between these scales makes it possible to fill in the gaps of models that consider only one of these scales.

The **regional scenarios** were quantified using the LuccME modeling framework [21]. LuccME is a spatially explicit dynamic modeling structure for LUCCs developed at the National Institute for Space Research (INPE). This approach makes it possible to delineate the spatial patterns of land use and land cover classes based on the components of (a) **Demand**, that is, the amount/intensity of changes in each use that is intended to be allocated over time [22, 23]; (b) **Potential**, which corresponds to the adequacy that a given cell in the cellular space has to change with each step of time, complete. In this case, using the spatial lag regression model [23–25] and (c) **Allocation** that spatially and interactively distributes the LUCC according to the previous components (demand and potential), based on competition between classes of land use in each cell. The land use and cover data used in the LuccMEBR were obtained from IBGE [26]. We chose this database because of its national scope, periodicity (2000, 2010, 2012, and 2014), and classes. In addition, IBGE data are consistent with other regional mappings such as TerraClass http://www.inpe.br/cra/projetos_pesquisas/dados_terraclass.php and MapBiomas https://plataforma.brasil.mapbiomas.org/.

For **global scenarios**, we use the projections generated by the Integrated Model to Assess the Global Environment (IMAGE) [27, 28], representing different combinations of SSPs and RCPs. The integration across scales to generate the scenarios is as follows: First, we defined the land change classes. Table 1 presents a description of the original and reclassified IBGE classes used for the LUCC modeling and the equivalent classes in the IMAGE model. The 13 classes from IBGE [26] were reclassified, similar grouping classes to reduce the complexity of the model. Second, we used the quantity of change projected by IMAGE for the selected combinations of SSPs/RCPs to define the quantity of change for each land use (LuccME demand component), as detailed in Section 2.3. Third, we developed local narratives related to the selected SSPs/RCPs. We parameterized the LuccME allocation and potential components using a comprehensive socioeconomic, institutional, and biophysical driver's database based on these narratives. Fourth, climate projection data for some of the RCPs were used to generate water deficit data using the integrated surface process model (INLAND) model. Fig 1 illustrates the integration/translation structure across scales to generate regionalized scenarios.

## 2.2 Scenario assumptions: From global to regional

The SSPs are based on five different development paths for societal trends (Table 2: i.e., sustainable development (SSP1), middle of the road developments (SSP2), global fragmentation (SSP3), strong inequality (SSP4), and rapid economic growth based on a fossil-fuel intensive energy system (SSP5). They were designed to represent different degrees of challenges in mitigation and adaptation. Each SSPs has been elaborated in a storyline and quantified using Integrated Assessment Models (IAM), such as IMAGE. The five SSP storylines can also be combined with alternative assumptions about climate mitigation, forming a matrix of alternative scenarios. Therefore, each SSP has a baseline scenario implementation and additional scenarios combining the storyline to climate mitigation policies compatible with certain levels of $CO_2$ concentration, and consequently, climate change. These mitigation assumptions are linked to RCPs, meaning the atmosphere's expected radiative forcing in 2100 (W m$^{-2}$). Each of the four RCPs has a different forcing at the end of the 21st century and is named according to its forcing level in 2100: RCP1.9 (1.9 W m$^{-2}$), RCP2.6 (3 W m$^{-2}$), RCP4.5 (4.5 W m$^{-2}$), RCP6.0 (6.0 W m$^{-2}$), RCP7.0 (7.0 W m$^{-2}$) and RCP8.5 (8.5 W m$^{-2}$). For instance, the RCP 2.6 assumes a 3 W m$^{-2}$ peak before 2100 and a decline to 2.6 W m$^{-2}$ by 2100.

**Table 1. Land use and land cover classes.**

| Original land use and cover classes—IBGE | LuccME/BR reclassification | Original land use and cover classes—IMAGE | Description (Source: IBGE [26] |
|---|---|---|---|
| Forest vegetation Humid Areas | Forest vegetation | Tropical woodland Tropical forest | Areas occupied by forests, which include areas of dense forest (forest structure with continuous top cover), Open Forest (forest structure with different degrees of discontinuity of the top cover, according to its type with vine, bamboo, palm, or sororoca), forest seasonal (forest structure with loss of leaves from the upper strata during the unfavorable season (dry and cold), in addition to the mixed rain forest (forest structure that comprises the natural distribution area of Araucaria angustifolia, a striking element in the upper strata, which generally forms a cover to be continued). In addition to these, other features were included, such as Savannah Forest, Campinarana Forest, Campinarana Arborizada, and Mangroves, as well as humid areas, which correspond to natural herbaceous vegetation, permanently or periodically flooded with fresh or brackish water (estuaries, swamps, etc.). In these areas, the land of ponds, swamps, and humid fields are inserted, among others. |
| Natural pasture | Grassland vegetation | Extensive grassland Scrubland Savana | The area is occupied by grassland vegetation subjected to grazing and other low-intensity anthropogenic interference. |
| Pasture planted | Pasture planted | Grassland-steppe | The area is predominantly occupied by cultivated herbaceous vegetation. They are places for cattle grazing, formed by planting perennial forages, subject to high-intensity anthropic interference, such as clearing the land (unblocking and cutting), liming, and fertilizing. |
| Agriculture | Agriculture | Agricultural land Biofuels | Areas occupied by temporary crops and permanent crops, irrigated or not, being land used for the production of food, fiber and agribusiness commodities. It includes all cultivated land, which may be planted or at rest, as well as a cultivated wetland. This can be represented by heterogeneous agricultural zones or extensive plantations. |
| Mosaic of agricultural area with forest remnants / Mosaic of forest vegetation with agricultural activity / Mosaic of agricultural areas with grassland remnants | Mosaic of occupation | Warm mixed forest | Mosaic of small-scale agricultural activities and remnants of natural vegetation (e.g., subsistence agriculture). |
| Forestry | Forestry | Regrowth forest timber | forests planted and managed with exotic species (e.g., eucalyptus and pine). |
| Artificial area / Bare land / Inland water bodies / Coastal water bodies | Others | Not considered | Artificial surfaces (e.g., cities, villages, transport routes, energy and communication networks) and bodies of water and bare lands (e.g., rock outcrops, dunes, cliffs). |

In our regionalized scenarios, we adopted the following combinations: SSP1 RCP 1.9, SSP2 RCP 4.5, and SSP3 RCP 7.0 (Table 3). In particular, the sustainable development scenario (SSP1) combined with stringent climate policy (RCP 1.9) is a scenario exploring the route toward a more sustainable world, providing an initial framework for our analysis of sustainability pathways. However, the Sustainable Development Goals (SDGs) have not been targeted in its development [20]. Mitigation scenarios that achieve the ambitious targets included in the Paris Agreement typically rely on greenhouse gas emission reductions combined with net carbon dioxide removal from the atmosphere, mostly accomplished through the large-scale application of bioenergy with carbon capture and storage, and afforestation (e.g., [16, 28]).

The premises of the regional scenario are based on the framework developed in the AMAZALERT project for the Brazilian Amazon [20], in line with the SSPs and RCPs described above. The scenarios range from low to high social development and high to low environmental development (Fig 2). We define high environmental development as the responsible management of natural resources (e.g., environmental stewardship), which includes high quality and equal access to services, opportunities, and resources supported by strong institutions.

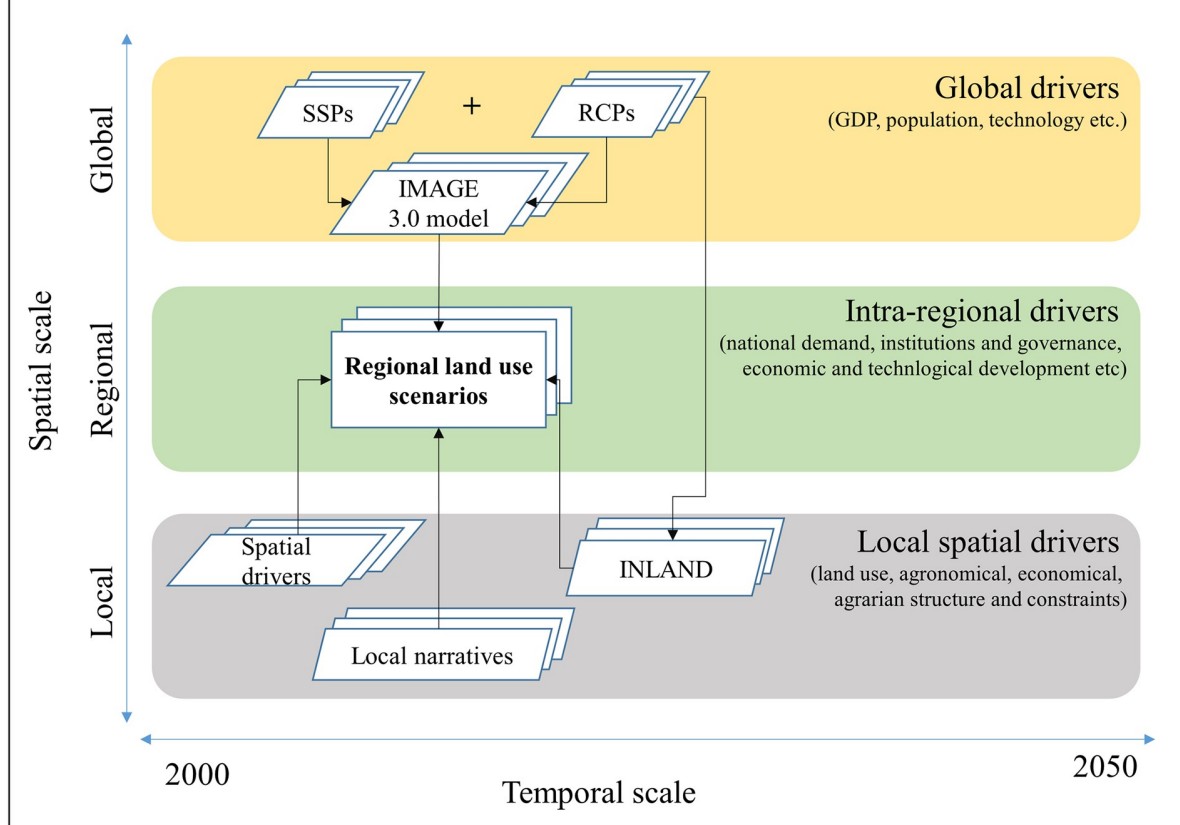

**Fig 1. Schematic representation of the development of regional land use and land cover scenarios.** Shared Socioeconomic Pathways (SSPs) and Representative Concentration Pathway (RCPs).

The sustainable development scenario (SSP1 RCP 1.9) assumes that all existing environmental laws are in place and policies to reduce deforestation, encourage environmental restoration, and preserve PAs and ILs. In this scenario, Brazil is gradually moving, like the world, toward a more sustainable path, emphasizing a more inclusive development that respects perceived environmental limits. Just as investments in education and health accelerate the demographic transition, the emphasis on economic growth human well-being. Inequality is reduced, and consumption is oriented toward low material growth, less resource and energy use, and healthier and less wasteful diets.

In the middle of the road scenario (SSP2 RCP 4.5), we assume that some of the positive trends in the last decade will be maintained. Still, they do not reach the full potential of an integrated socioeconomic, institutional, and environmental perspective. In this context, conservation, agricultural and extractive policies and initiatives continue to be a source of tension and contradiction. Forest governance remains centralized, with the national government playing an important role in decision-making. In line with global challenges, development and income growth occur unevenly, despite some institutions working to achieve SDGs. Despite some improvements and advances in combating this problem, degradation of environmental systems is still present. Regarding the use of resources and energy, there was a slight decrease, and population growth remained moderate.

The strong inequality scenario (SSP3 RCP 7.0) reflects a weakening of efforts in recent years, mainly in the socio-environmental dimension. As in the rest of the world, nationalism is

**Table 2. Synthesis of core premises differentiating the SSPs in relation to land use.** Source: Popp et al. [29].

| | SSP 1 | SSP 2 | SSP 3 | SSP 4 | SSP 5 |
|---|---|---|---|---|---|
| Land-use change regulation | Strong regulation to avoid environmental tradeoffs | Medium regulation; slow decline in the rate of deforestation | Limited regulation; continued deforestation | Highly regulated in medium-income (MICs) and high-income (HICs) countries; lack of regulation in low-income countries (LICs) lead to high deforestation rates | Medium regulation, slow decline in the rate of deforestation |
| Land productivity growth | High improvements in agricultural productivity; Rapid diffusion of best practices | Medium pace of technological change | Low technology development | High productivity for large-scale industrial farming, low for small-scale farming | Highly managed, resource-intensive; rapid increase in productivity |
| Environmental Impact of Food Consumption | Low Growth in Food Consumption, Low-Meat Diets | Material-Intensive Consumption, Medium Meat Consumption | Resource-Intensive Consumption | Elites: High-consumption Lifestyles; Rest: Low Consumption | Material-Intensive Consumption |
| International Trade | Moderate | Moderate | Strongly constrained | Moderate | High, with regional specialization in production |
| Globalization | Connected markets, regional production | Semi-open globalized economy | De-globalizing, regional security | Globally connected elites | Strongly globalized |
| Land-based mitigation policies | No delay in international cooperation for climate change mitigation Full participation of the land-use sector | Delayed international cooperation for climate change mitigation Partial participation of the land-use sector | Heavily delayed international cooperation for climate change mitigation. Limited participation of the land-use sector | No delay in international cooperation for climate change mitigation Partial participation of the land-use sector | Delayed international cooperation for climate change mitigation Full participation in the land-use sector |

resurgent, concerns about competitiveness and security, and regional conflicts. Policies change over time to become increasingly oriented toward national and regional security issues. Countries focus on achieving energy and food security goals in their regions at the expense of broader-based development. Investments in education and technological developments have

**Table 3. Detailed regional assumptions related to Sustainable development scenario, middle of the road, and strong inequality scenario.**

| Quantification element | | Sustainable development scenario SSP1 RCP 1.9 | Middle of the road scenario SSP2 RCP 4.5 | Strong inequality scenario SSP3 RCP 7.0 |
|---|---|---|---|---|
| (a) (Environmental) Law enforcement | | Forest Code Restoration (Legal reserves (LR) and Permanent protection areas (PPA)) and Conservation measures are enforced, incentivized, and even surpassed, promoting a Forest Transition process 2030. Protected areas are fully implemented and respected. | Forest Code Restoration (Legal reserves (LR) and Permanent protection areas (PPA)) measures are satisfied by compensation mechanisms, such as remote forest quotas, instead of local restoration. Forest code conservation measures are respected, and deforestation control mechanisms occur. | Forest Code is not respected, and deforestation control measures are discontinued. Protected areas were not fully implemented or protected. |
| (b) Changes in spatial drivers | Roads network | No major federal or state roads were built after 2020. | Same as Scenario C, but accompanied with measures to avoid uncontrolled occupation. | Ongoing paving concluded in 2025 (BR-163, BR-319, and BR-230). All paving and planned roads (Federal and State) were built and distributed by 2025, 2030, and 2040, respectively. |
| | Rural settlements | Existing settlements are maintained, and non-conventional ones (sustainable) are well protected. | Existing settlements are maintained, but the non-conventional ones (sustainable) are less protected than Scenario A. | Rural settlements are canceled, and their areas become private property. |
| | Protected areas—PAs | Maintenance of the 2016 protected areas network. Fully protected. | Same as Scenario A in terms of area, but less protected in more densely occupied areas. | Decrease in the extension and level of protection of UCs by 2030. Maintenance of Regularized and Approved Indigenous Land |

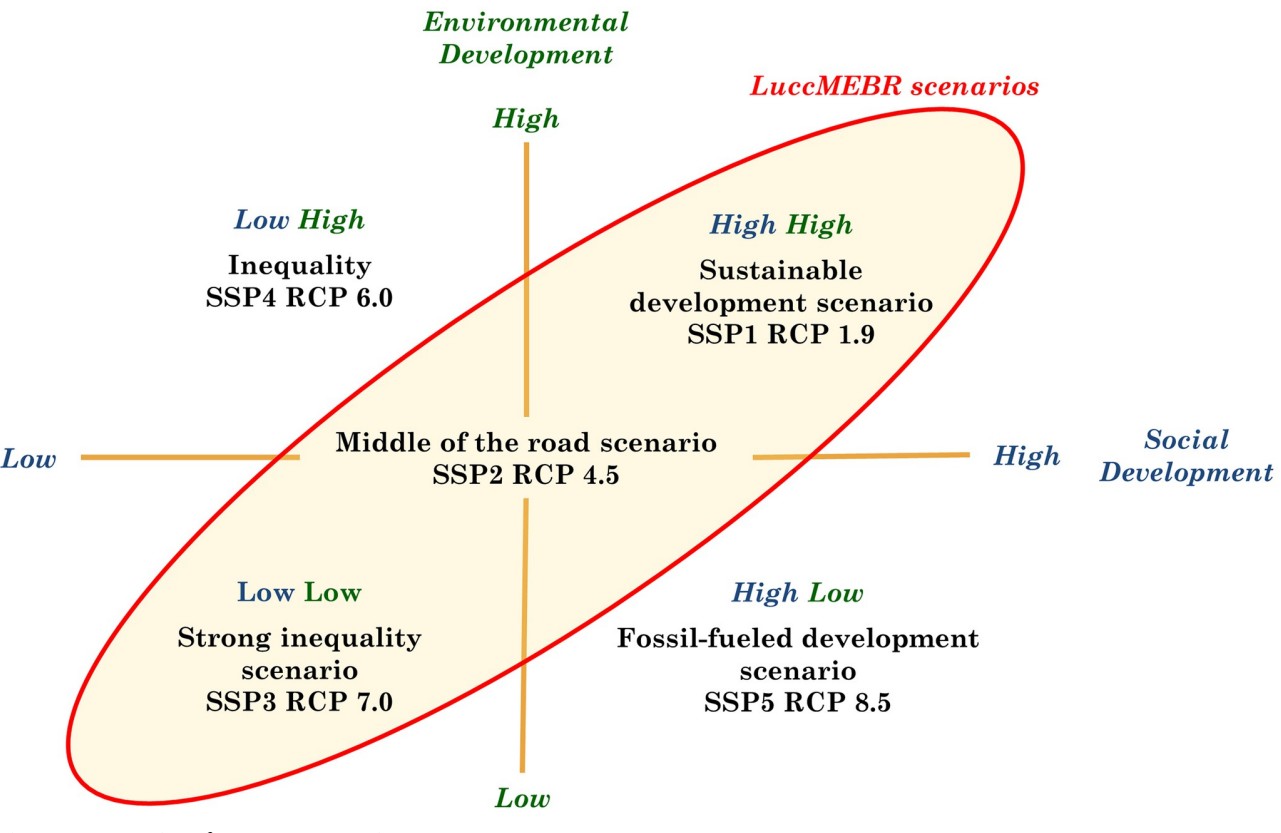

**Fig 2. Representation of LuccMEBR scenarios.**

decreased. Economic development is slow, and consumption is material-intensive. In this context, inequalities persist or worsen over time. Population tends to increase and environmental degradation intensifies mainly because of the low national and international priorities to address environmental issues.

## 2.3 LuccME model parameterization and validation

**2.3.1 Demand component.**   We use the LuccME *PreComputedValues* component, in which we externally calculate demand and report the expected area for each land-use class annually from 2000 to 2050 (Table 4). As described above, we used the amount of change projected by IMAGE. We adjusted it to the IBGE land use and land cover classes for the SSP1/RCP1.9, SSP2/RCP4.5, and SSP3/RCP7.0 combinations to generate the annual demand for each land-use class in each scenario between 2015 and 2050.

Eq 1 presents the calculation of the annual change $C_{ca}$ for each class of land use and land cover in the area unit.

$$C_{ca} = \frac{L_{ct_f} - L_{ct_i}}{n_t}, \tag{1}$$

where $C_{ca}$ corresponds to the annual change in area of the land use class $L_c$ between the initial $t_i$ and $t_f$ end year of the chosen period, and $n_t$ refers to the number of years in the period.

The calculation of the annual demand $D_{cat_k}$ of the land use class is represented by Eq 2:

$$D_{cat_k} = L_{ct_k-1} + C_{ca}, \tag{2}$$

**Table 4. Land demand parameters.**

| PreComputed values | | SSP1 RCP 1.9 | | SSP2 RCP 4.5 | | SSP3 RCP 7.0 | |
|---|---|---|---|---|---|---|---|
| Forest vegetation | *from* | 3,771,677 | km $^2$ (2000) | 3,771,677 | km $^2$ (2000) | 3,771,677 | km $^2$ (2000) |
| | *to* | 4,232,402 | km $^2$ (2050) | 2,975,314 | km $^2$ (2050) | 2,846,765 | km $^2$ (2050) |
| Grassland vegetation | *from* | 2,112,947 | km $^2$ (2000) | 2,112,947 | km $^2$ (2000) | 2,112,947 | km $^2$ (2000) |
| | *to* | 1,517,633 | km $^2$ (2050) | 1,012,637 | km $^2$ (2050) | 772,878 | km $^2$ (2050) |
| Pasture planted | *from* | 402,529 | km $^2$ (2000) | 402,529 | km $^2$ (2000) | 402,529 | km $^2$ (2000) |
| | *to* | 445,974 | km $^2$ (2050) | 670,850 | km $^2$ (2050) | 714,860 | km $^2$ (2050) |
| Agricultural area | *from* | 624,941 | km $^2$ (2000) | 624,941 | km $^2$ (2000) | 624,941 | km $^2$ (2000) |
| | *to* | 328,444 | km $^2$ (2050) | 856,018 | km $^2$ (2050) | 1,077,091 | km $^2$ (2050) |
| Mosaic of occupations | *from* | 1,405,301 | km $^2$ (2000) | 1,405,301 | km $^2$ (2000) | 1,405,301 | km $^2$ (2000) |
| | *to* | 1,822,044 | km $^2$ (2050) | 2,788,498 | km $^2$ (2050) | 2,873,629 | km $^2$ (2050) |
| Forestry | *from* | 55,983 | km $^2$ (2000) | 55,983 | km $^2$ (2000) | 55,983 | km $^2$ (2000) |
| | *to* | 26,881 | km $^2$ (2050) | 70,060 | km $^2$ (2050) | 88,154 | km $^2$ (2050) |

where $D_{cat_k}$ corresponds to the annual demand of a given land use class $L_c$ in a given year $t_k$, calculated from the sum of the class area in the previous year $t_{k-1}$, and the annual change $C_{ca}$.

In the initial year, the value of the demand corresponds to the observed amount of the land-use class, calculated based on the use and land cover data used; in this case, the land use and land cover change data from IBGE [26].

**2.3.2 Potential and allocation component parametrization: Intraregional and local spatial drivers.** The LuccME component is used to determine the potential occurrence of a given land use cover class, as well as the ***PotentialCSpatialLagRegression*** (Eq 3), which is based on and adapted from the spatial regression model (spatial lag) [23–25]. In this component, the influence of neighboring areas is considered to occur. This is an intrinsic feature of land use and land cover changes. In addition, this component allows this potential to be dynamic over the modeled period, that is, every year.

$$\mathbf{Pot}_{cxyt} = \%\boldsymbol{RegL}_{cxyt} - \%\boldsymbol{L}_{cxyt-1} \;\; : \;\; \{\boldsymbol{Pot}_{cxyt} \in \Re \;\|\; -1 \leq \boldsymbol{Pot}_{cxyt} \leq 1\}, \tag{3}$$

where $\boldsymbol{Pot}_{cxyt}$ corresponds to the potential for the occurrence of a given land use class $L_c$ in a given location $\boldsymbol{xy}$ in a given time step $\boldsymbol{t}$. To determine the potential, the percentage of use estimated by the regression $\boldsymbol{RegL}_{cxyt}$ is subtracted from the percentage of existing use $\boldsymbol{L}_{cxy}$ at time $\boldsymbol{t\text{-}1}$.

To calculate the potential, the variables potentially explaining the process of changing land use and coverage in different Brazilian biomes were divided into four categories: agronomic aspects (composed of climatological and geophysical variables), agrarian structure (percentage of the area of agricultural establishments), economic aspects (dependent on structural variables: distance from highways, ports, airports, railways, urban centers, rivers, etc.), and restrictive aspects (related to legal limitations: protected areas, conservation units, rural settlements, distance to hydroelectric and thermoelectric in operation). All candidate variables were integrated into regular 10 km × 10 km cells for the spatially explicit analysis, taking as base years 2000 and 2010 (Fig 3). For this, the ***FillCell plugin*** was used [30]. The use of cellular space made it possible to homogenize the factors described above, regardless of their origin format (vector data, matrix data, etc.), aggregating them on the same space-time basis, through operators (e.g., percentage of each class, minimum distance, etc.) used according to the geometric representation and the semantics of the attributes of the input data.

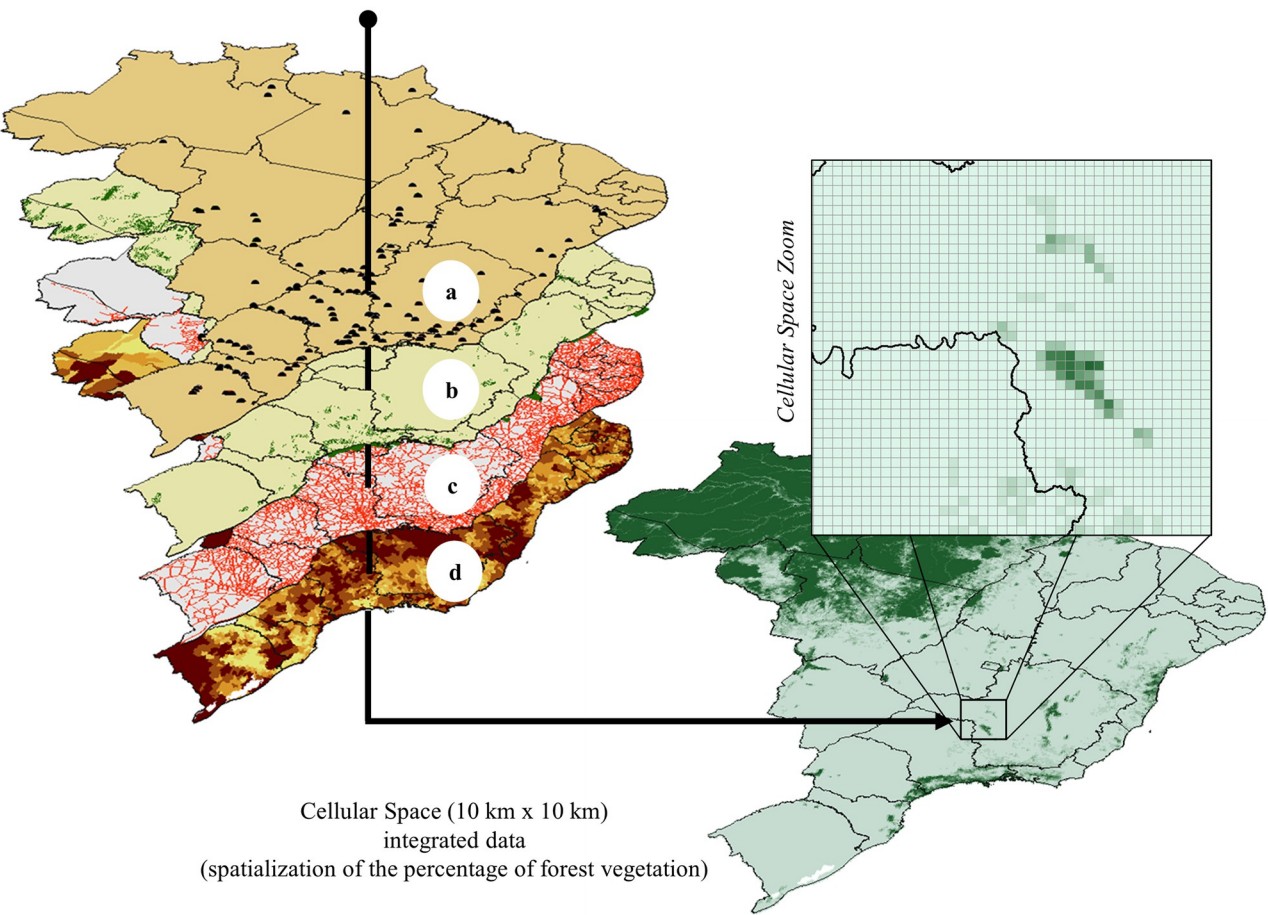

Cellular Space (10 km x 10 km)
integrated data
(spatialization of the percentage of forest vegetation)

**Fig 3. Representation of the factors integration into the cellular space.** a) Hydroelectric plants, b) Protected areas, c) Federal and State highways, and d) Proportion of large agricultural establishments. The source of States boundaries is according to IBGE [31].

Searching for a model that involves the minimum of possible parameters to be estimated and explains well the behavior of land use in each Brazilian biome, some statistical techniques were used to evaluate the effectiveness and adequacy of the best model: a priori, Spearman's correlation analysis selected only those factors that presented a correlation coefficient below or equal to 0.60; second, with the new composition of candidate variables, a spatial regression analysis was performed (Spatial Lag [24, 25]), considering the determination coefficient ($R^2 >$ 0.75, in the average of uses) as decision parameters; statistical significance (p-value < 0.05) and Akaike information criterion (AIC) (lowest values obtained). The calibration of the model was carried out using observational data contained in the land use and land cover maps of IBGE in 2000, 2010, 2012, and 2014.

The allocation of land-use change in each scenario was based on the application of the LuccME *AllocationCClueLike* allocation component based on the CLUE [32], which was distributed spatially and interactively according to the previous components (demand and potential), based on competition between the types of land use in each cell and within a previously established maximum error. For the parameterization of this component, the Forest Code rules regarding the amount of legal reserve required according to the regions of the Brazilian territory were also considered. The allocation process for each type of land use or land cover

can be described using Eq 4.

$$L_{cxyt} = L_{cxyt-1} + Pot_{cxyt} * ITF_c, \quad (4)$$

where the amount of area allocated from a given class of land use $L_c$ at a given $xy$ location in the cell plane at time $t$ is determined in an iterative process of the sum of $L_{cxy}$ at time $t\text{-}1$ and the potential $Pot_{cxyt}$ multiplied by an adjustment factor proportional to the difference between the allocated area, the reported demand, and the direction of the change $ITF_c$.

The parameterization details according to potential (*PotentialCSpatialLagRegression*), Allocation (*AllocationCClueLike*), and demand (*DemandPreComputedValues*) components are presented in the supplementary material S1 and S2 Appendices.

**2.3.3 Model validation.** The results of the simulations were validated by the multiresolution adjustment validation metric, adapted from Costanza [33], and Pontius Jr [34]. This metric allows establishing the level of similarity between the simulated and real maps at different resolutions through sampling windows that increase with each step of time; therefore, this approach allows the evaluation of both location errors in the resolution of the model itself and spatial pattern errors, degrading the resolution of maps. This metric is particularly useful for characterizing land use and land cover change and for validating land use, and land cover change models [34]. The similarity level can be calculated according to Eq 5:

$$NSi = 1 - \left[ \frac{\sum_{j=1}^{n} (| \sum_{c=1}^{k} dif_{sim,c} - \sum_{c=1}^{k} dif_{real,c} |)}{2 * \sum_{j=1}^{n} \sum_{c=1}^{k} dif_{real,c}} \right] * 100, \quad (5)$$

where $NS$ corresponds to the level of similarity between the real and simulated maps at a given resolution $i$; $j$ is the window/cells considered; $n$ establishes the number of windows/cells to be considered; textit**c** is the number of cells in a resolution $k$ $(i^*i)$; and $dif_{real}$ = % real$_{tf}$ − % real$_{ti}$ and $dif_{sim}$ = % sim$_{tfinal}$ − % real$_{initial}$ being $ti$ and $tf$ the initial and real years, respectively, considered in the validation.

# 3 Data records

The dataset provides 35 maps of the spatial distribution of land use and coverage for Brazil between 2000 and 2050, considering the scenarios discussed above. The data are made available in the SIRGAS2000 Polychronic Projection System with Datum (EPSG 5880). The file is available in Zenodo (https://zenodo.org/record/5123560) [35] in NetCDF format. The data set contains maps with the percentage of forest vegetation, grassland vegetation, managed pasture, agriculture, mosaic of occupation, and forestry in 10 km × 10 km cells.

# 4 Model performance

Figs 4 and 5 present the spatial distribution of the use classes and the dissimilarity between the observed and simulated data in the validation year 2014. The performance of the LuccMEBR model was satisfactory, with an average spatial fit index, between the observed and simulated data in 2014, of 84.44% when comparing the patterns of both maps (Table 5). When considering only the areas where some change occurred, the adjustment index was 52.55% for each land-use type. The average percentage of adjustment errors corresponding to omissions was 6.80%, while commission errors were approximately 6.39%. Among the land use and land cover classes, the highest general spatial adjustment values were observed for Forest vegetation, Grassland vegetation, Agriculture, and Planted pasture.

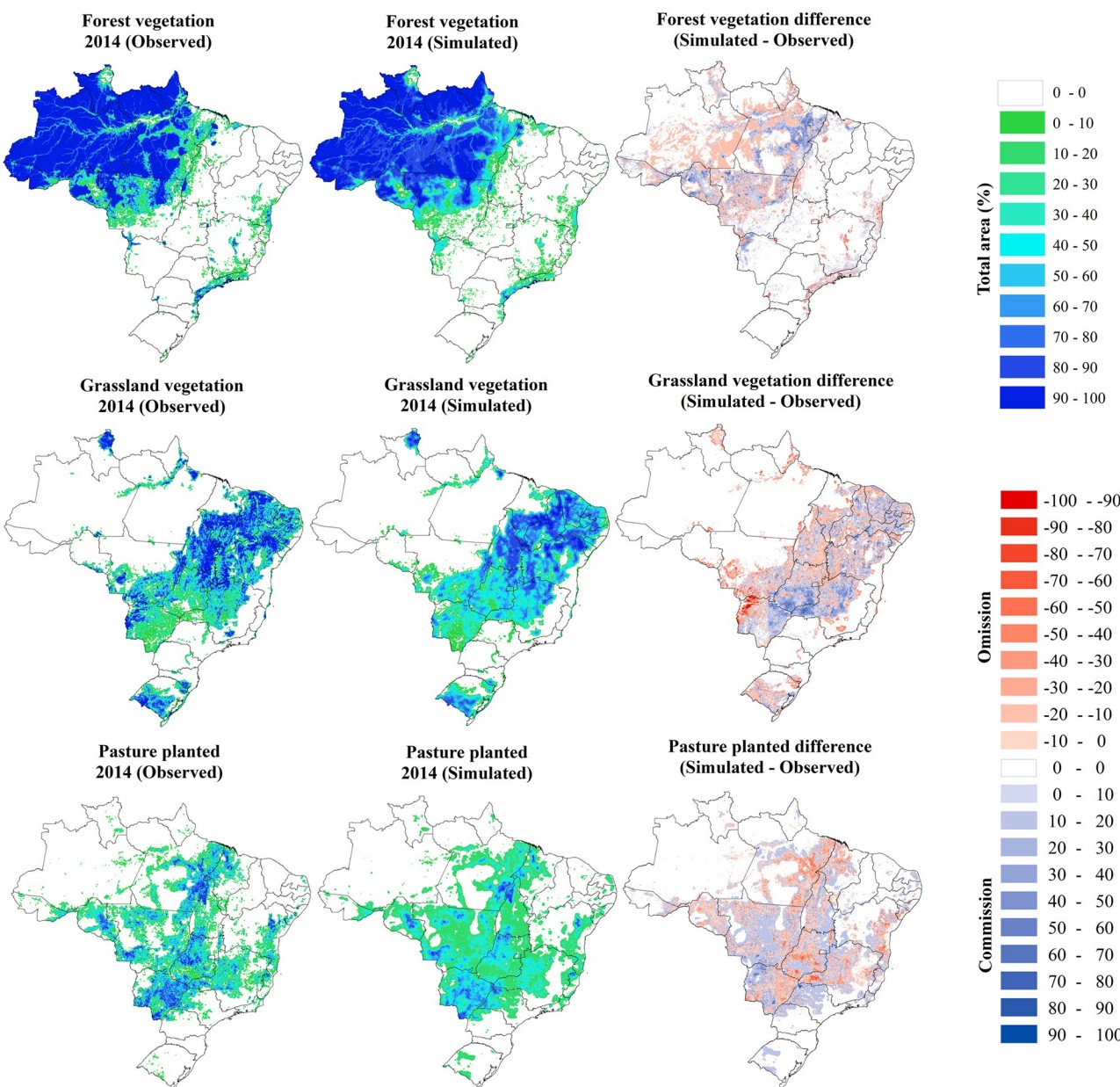

**Fig 4. Percentage of forest vegetation, grassland vegetation and pasture planted observed versus simulated in 10 x 10 km² cells in 2014 and the spatial distribution of errors of omission and commission.** The source of States boundaries is according to IBGE [31].

## 5 Usage notes

The data on land use and coverage presented for the entire Brazilian territory comes from an effort to align the development of regional scenarios with the structure of global scenarios (RCP–SSP–SPA) were developed by the IPCC. Despite the important challenges that this alignment adds to the development of these scenarios, which are: (i) the additional complexity in capturing the multiple dimensions of change and (ii) the issues of scale [36], the results obtained from this process have greater consistency between the different spatial and temporal scales of interest. In addition, the development and study of regional scenarios help

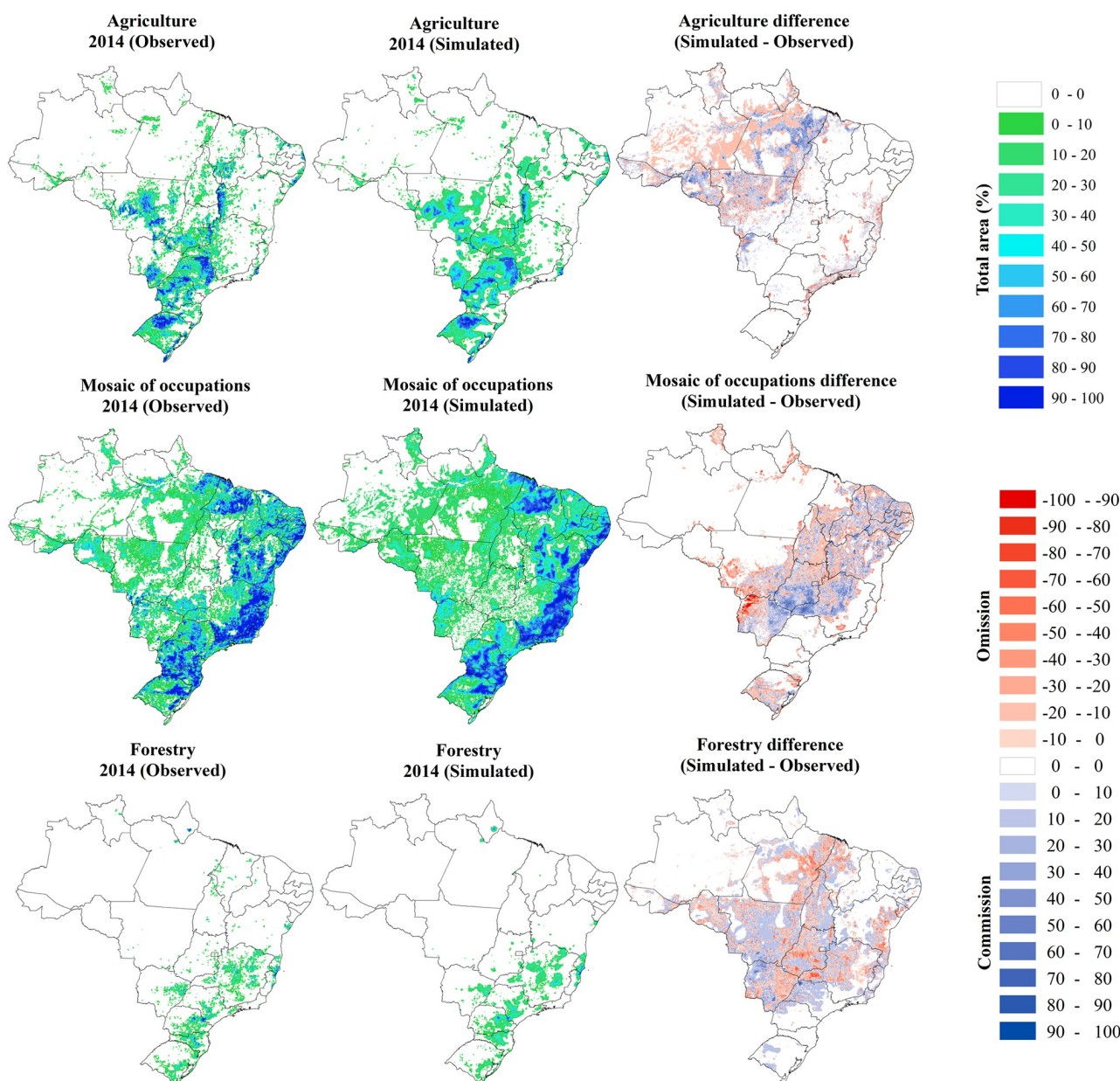

**Fig 5. Percentage of Agriculture, Mosaic of occupation and Forestry observed versus simulated in 10 x 10 km² cells in 2014 and the spatial distribution of errors of omission and commission.** The source of States boundaries is according to IBGE [31].

policymakers and the scientific community to develop robust strategies in the face of uncertain futures and evaluate and improve the feasibility, flexibility, and concreteness of their actions [37–41]. Fig 6 shows the spatially explicit distribution of the classes of use in the initial year of the simulation (2000) and the three scenarios considered. It can be seen that the middle of the road and strong inequality scenarios present similar patterns in all classes of use, with emphasis on the significant increase in the mosaic class of occupations, with intensification in the Caatinga and Atlantic Rainforest. Unlike the other scenarios, the regeneration of forest vegetation was observed in the sustainable development scenario, mostly in the Caatinga, Atlantic Rainforest, Cerrado, and Pampa biomes.

**Table 5. Percentage of spatial adjustment and errors.**

| | Spatial adjustment | | Errors | |
|---|---|---|---|---|
| | Patterns | Modified areas | Omissions | Commission |
| | % | | | |
| Forest vegetation | 95.45 | 58.91 | 7.03 | 5.83 |
| Grassland vegetation | 87.24 | 61.82 | 11.42 | 9.03 |
| Planted pasture | 81.75 | 56.46 | 6.63 | 7.97 |
| Agriculture | 85.28 | 53.22 | 4.53 | 4.09 |
| Mosaic of occupation | 82.42 | 39.27 | 7.23 | 8.57 |
| Forestry | 74.48 | 45.63 | 3.94 | 2.83 |
| Average | 84.44 | 52.55 | 6.80 | 6.39 |

Analyzing the dynamics of LUCC (Table 6), according to the scenarios considered, has been observed; agriculture, the mosaic of occupation, as well as grassland vegetation will continue in the same direction, regardless of the scenario considered. Concerning the other classes, it can be seen that the sustainable development scenario is distinguished from the others, as well as in the spatial pattern observed in Fig 6.

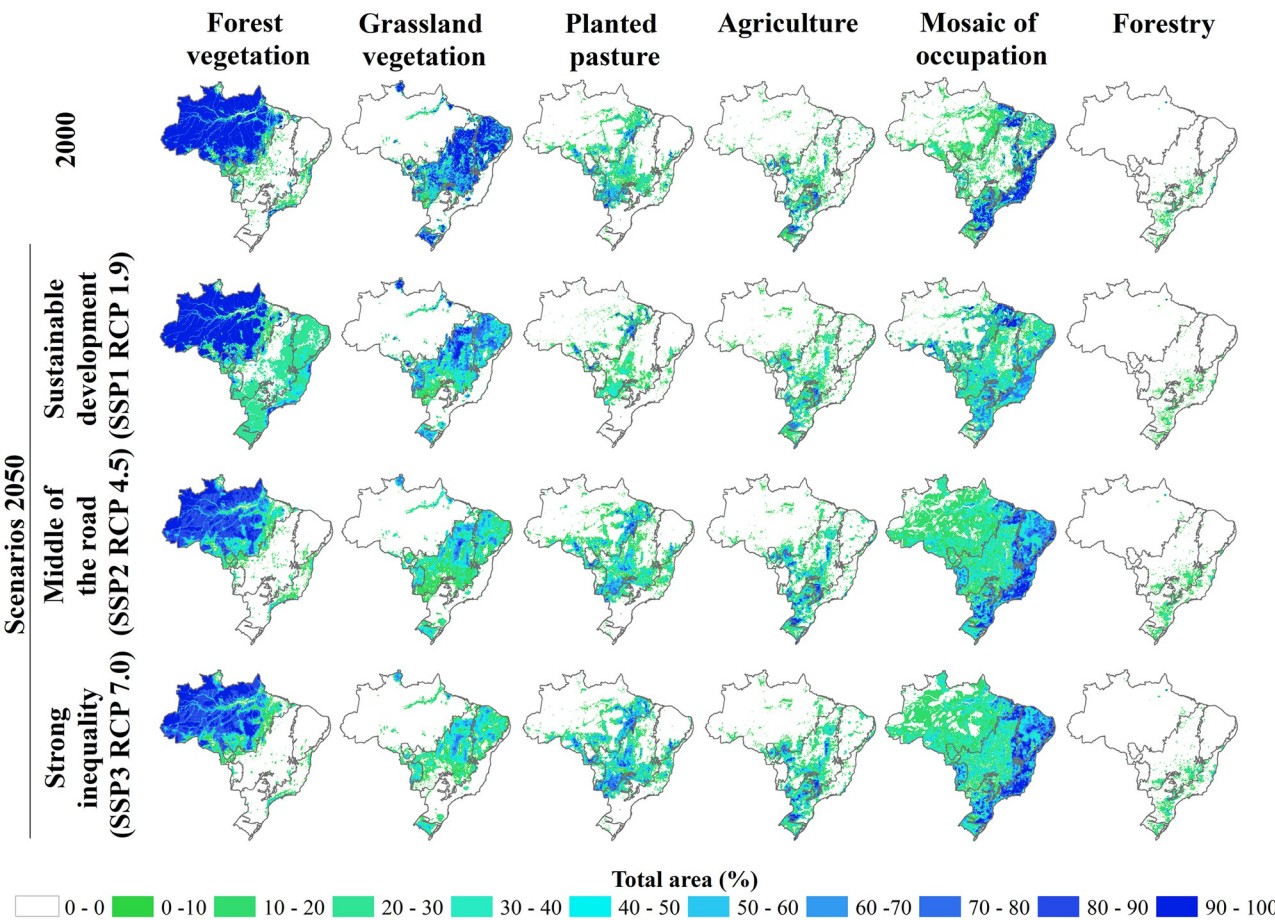

**Fig 6. Spatial distribution of areas and land use according to the scenarios from 2000 to 2050.** The source of States boundaries is according to IBGE [31].

**Table 6. The direction of change in land use and coverage, according to classes and scenarios between 2000 and 2050.** ↗ = Increase and ↘ = Reduction.

|  | Forest vegetation | Grassland vegetation | Planted pasture | Agriculture | Mosaic of occupation | Forestry |
|---|---|---|---|---|---|---|
| **Sustainable development scenario SSP1 RCP 1.9** | ↗ | ↘ | ↘ | ↗ | ↗ | ↘ |
| **Middle of the road scenario SSP2 RCP 4.5** | ↘ | ↘ | ↗ | ↗ | ↗ | ↗ |
| **Strong inequality scenario SSP3 RCP 7.0** | ↘ | ↘ | ↗ | ↗ | ↗ | ↗ |

According to the middle of the road and strong inequality scenarios, forest vegetation will suffer a reduction of approximately 805,956 and 933,092 km$^2$, respectively, until 2050, mainly in the Amazon biome (673,066 and 762,739 km$^2$), followed by the Cerrado biome (67,643 and 87,288 km$^2$). However, in the sustainable development scenario, forest vegetation will increase, occupying 447,944 km$^2$. This increase occurs mostly in the Atlantic Forest (253,783 km$^2$), Cerrado (200,489 km$^2$), and Caatinga (168,572 km$^2$) biomes, as shown in Fig 6. However, the Amazon biome will be reduced by 217,696 km$^2$ of forest vegetation, approximately 2/3 less than the values observed in the other scenarios. As shown in Fig 6 and Table 6, there is a reduction in grassland vegetation areas in both scenarios. Overall, in the strong inequality scenario, the reduction will be approximately 1,346,988 km$^2$, followed by the middle of the road scenario with 1,107,235 km$^2$ and the sustainable development scenario with 602,847 km$^2$. Equivalent to what will occur with Forest vegetation, this reduction will occur mainly in the Cerrado (586,575 km$^2$, on average) and Caatinga (303,419 km$^2$, on average) biomes. In an inverse to what will happen with forest vegetation, planted pasture and forestry will tend to increase their extension in the middle of the road and strong inequality scenarios (219,199 and 439,962 km$^2$, respectively). In contrast, in the sustainable development scenario, there will be a reduction of approximately 308.206 km$^2$. Despite the data showing an increase in agriculture and mosaic of occupation, regardless of the scenario considered, the increase will occur with greater intensity in the middle of the road and strong inequality scenarios, whereas in agriculture, the increase will be 259,892 and 303,781 km$^2$, respectively; in the mosaic of occupation, this increase will be 1,366,687 and 1,450,867 km$^2$, respectively. Although smaller, the increase in the sustainable development scenario will correspond to 34,973 km$^2$ in agriculture and 403,914 km$^2$ in the mosaic of occupation. It should be noted that the increase in the areas of planted pasture, forestry, and agriculture should occur in the Cerrado biome. In contrast, the mosaic of occupation will increase, mostly in the Amazon and Cerrado biomes.

This set of scenarios provides important information that can help establish public policies that aim to contribute to biodiversity conservation and reduce emissions from deforestation and degradation, especially those arising from land use and coverage changes. Furthermore, this set of scenarios with territorial extension for the whole of Brazil makes it possible to understand how decision-making and global demands can influence other regions.

The study demonstrated the benefits of a multidimensional scenario framework for integrating different land-use factors and land cover change. The inclusion of participatory methods that support the elaboration of participatory scenarios, with qualitative and quantitative components, can enrich the scenarios presented.

## Supporting information

**S1 Appendix. LuccMEBR: Scenario-dependent spatiotemporal drivers.**
(PDF)

**S2 Appendix. LuccMEBR: Model parameters.**
(XLSX)

## Acknowledgments

The authors thank Eloi Dalla Nora and Detlef Van Vuuren for their contributions to developing the scenarios.

## Author Contributions

**Conceptualization:** Francisco Gilney Silva Bezerra, Ana Paula Aguiar.

**Data curation:** Francisco Gilney Silva Bezerra, Ana Paula Aguiar.

**Formal analysis:** Francisco Gilney Silva Bezerra, Celso Von Randow.

**Funding acquisition:** Ana Paula Aguiar.

**Investigation:** Francisco Gilney Silva Bezerra, Karine Rocha Aguiar Bezerra, Graciela Tejada, Aline Anderson Castro.

**Methodology:** Francisco Gilney Silva Bezerra, Talita Oliveira Assis, Ana Paula Aguiar.

**Project administration:** Celso Von Randow, Ana Paula Aguiar.

**Software:** Talita Oliveira Assis, Diego Melo de Paula Gomes, Rodrigo Avancini.

**Supervision:** Celso Von Randow, Ana Paula Aguiar.

**Validation:** Francisco Gilney Silva Bezerra, Karine Rocha Aguiar Bezerra, Graciela Tejada, Aline Anderson Castro.

**Writing – original draft:** Francisco Gilney Silva Bezerra, Celso Von Randow, Talita Oliveira Assis, Karine Rocha Aguiar Bezerra, Graciela Tejada, Aline Anderson Castro, Diego Melo de Paula Gomes, Rodrigo Avancini, Ana Paula Aguiar.

**Writing – review & editing:** Francisco Gilney Silva Bezerra, Celso Von Randow, Talita Oliveira Assis, Karine Rocha Aguiar Bezerra, Graciela Tejada, Aline Anderson Castro, Diego Melo de Paula Gomes, Rodrigo Avancini, Ana Paula Aguiar.

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
