## [Decision Letter · Decision Letter 0]

5 Oct 2021

PONE-D-21-24083New land use change scenarios for Brazil: refining global SSPs with a regional spatially-explicit allocation modelPLOS ONE

Dear Dr. Silva Bezerra,

Thank you for submitting your manuscript to PLOS ONE. After careful consideration, we feel that it has merit but does not fully meet PLOS ONE’s publication criteria as it currently stands. Therefore, we invite you to submit a revised version of the manuscript that addresses the points raised during the review process.

Both reviewers agreed that the paper is well organised and written. However, there are both minor and major issues highlighted by the reviewers to be revised to improve the manuscript. 

We look forward to receiving your revised manuscript.

Kind regards,

Eda Ustaoglu, PhD

Academic Editor

PLOS ONE

Journal Requirements:

4. We note that Figures 2, 3 and 4 in your submission contain [map/satellite] images which may be copyrighted. All PLOS content is published under the Creative Commons Attribution License (CC BY 4.0), which means that the manuscript, images, and Supporting Information files will be freely available online, and any third party is permitted to access, download, copy, distribute, and use these materials in any way, even commercially, with proper attribution. For these reasons, we cannot publish previously copyrighted maps or satellite images created using proprietary data, such as Google software (Google Maps, Street View, and Earth). For more information, see our copyright guidelines: http://journals.plos.org/plosone/s/licenses-and-copyright.

a. You may seek permission from the original copyright holder of Figures 2, 3 and 4 to publish the content specifically under the CC BY 4.0 license.  

Reviewers' comments:

Reviewer's Responses to Questions

**Comments to the Author**

1. Is the manuscript technically sound, and do the data support the conclusions?

Reviewer #1: Yes

Reviewer #2: Partly

2. Has the statistical analysis been performed appropriately and rigorously? 

Reviewer #1: Yes

Reviewer #2: I Don't Know

3. Have the authors made all data underlying the findings in their manuscript fully available?

Reviewer #1: Yes

Reviewer #2: Yes

4. Is the manuscript presented in an intelligible fashion and written in standard English?

Reviewer #1: Yes

Reviewer #2: Yes

5. Review Comments to the Author

Reviewer #1: The paper entitled as "New land use change scenarios for Brazil: refining global SSPs with a regional spatially-explicit allocation model" present a downscaled spatial model for estimating future of land use and cover change in Brazil. The authors have used both global and regional parameters to estimate and train the transition models for projecting the LULC data until 2050.

The paper is well written and easy to follow, while the authors have provided many details regarding their data, models used and presenting intermediate results. The paper can be accepted for publication, I would only recommend to the authors to address some of the points given below:

1. Please elaborate further the Model performance and the results obtained in Figure 3. As the overall data are projections the validation with existing and simulated data remains a critical parameter that needs to be discussed. If possible please provide more details (per class) and more statistics in this section (probably using a table).

2. 2.2 Scenario assumptions: from global to regional and especially the paragraph after line 92. Please provide more details of how you have elaborated the downscaling approach moving from global to regional approach. Why and how did you selected the “ SSP1 RCP 1.9, SSP2 RCP 4.5 and SSP3 RCP 7.0—table 3). This is something that needs to be further elaborated and discussed.

3. Future work and research direction from the authors would be appreciated. What is missing in the current study, are there any steps for the future in terms of improvements of the model

Overall this is an interesting work.

Reviewer #2: This is an important topic. This study develops new land use change scenarios by including the global structure and local factors. The paper is well written and easy to follow.

Some comments:

1. Please add the definitions of “global”, “local”, and “balance” here. Take the study area, Brazil, as our example, does the global scale here mean real “global”, America, or Latin America? Giving examples of global or local factor can help our readers understand this research problems and focus on the key issues.

2. The authors can add explanations of their scenarios in the abstract. For example, sustainable development scenario for SPP1 RCP 1.9, middle of the road scenario for SPP2 RCP 4.5, and strong inequality scenario for SPP3 RCP 7.0. Also, what is the number meaning after RCP?

3. This study includes many abbreviations, such as LUCC, LuccMEBR, IBGE, IMAGE, LuccME, INLAND, LR, and PPAS. The authors can provide a list in the appendix.

4. There is a reference citation error on line 123 page 6.

5. In table one, the description of some classes are very detailed, but some are more generalized. Is there a specific reason?

6. For the study method, the authors used a validation matric (equation 5) to test the similarity between the real and simulated maps. Is this the most common method because the reference paper is 1989. Also, what is the range of this NS index and how to explain it?

7. Please increase resolution of figure 3 and 4.

6. PLOS authors have the option to publish the peer review history of their article (what does this mean?). If published, this will include your full peer review and any attached files.

Reviewer #1: No

Reviewer #2: No

---

## [Author Response · Author response to Decision Letter 0]

30 Dec 2021

Reviewer n. 1

Question: 1 . Please elaborate further the Model performance and the results obtained in Figure 3. As the overall data are projections the validation with existing and simulated data remains a critical parameter that needs to be discussed. If possible please provide more details (per class) and more statistics in this section (probably using a table).

Answer: We appreciate the suggestion. We inserted a table (Table 5) quantifying the performance for each land use and land cover class.

Question: 2. 2.2 Scenario assumptions: from global to regional, and especially the paragraph after line 92. Please provide more details of how you have elaborated the downscaling approach moving from global to regional approach. Why and how did you selected the “ SSP1 RCP 1.9, SSP2 RCP 4.5 and SSP3 RCP 7.0—table 3). This is something that needs to be further elaborated and discussed.

Answer: We appreciate the observation and suggestion. We have inserted the storylines for clarity. Line 106-152

Question: 3. Future work and research direction from the authors would be appreciated. What is missing in the current study, are there any steps for the future in terms of improvements of the model.

Answer: Thanks for the suggestion. We have included a paragraph (L318-321 ) indicating the importance of developing participatory scenarios in the future.

Reviewer n. 2

Question: 1. Please add the definitions of “global”, “local”, and “balance” here. Take the study area, Brazil, as our example, does the global scale here mean real “global”, America, or Latin America? Giving examples of global or local factor can help our readers understand this research problems and focus on the key issues.

Answer: Thanks for the suggestion. We have included a paragraph (L47-57) defining the three scales considered in developing the regional scenarios along with examples of factors/drivers.

Question: 2. The authors can add explanations of their scenarios in the abstract. For example, sustainable development scenario for SPP1 RCP 1.9, middle of the road scenario for SPP2 RCP 4.5, and strong inequality scenario for SPP3 RCP 7.0. Also, what is the number meaning after RCP?

Answer: Thanks for the suggestion. The numbers refer to the forcing for each RCP, the range of radiative forcing values in the year 2100. Radiative forcing is the change in the net, downward minus upward, radiative flux (expressed in watts per square meter; W m-2) at the tropopause or top of atmosphere due to a change in an external driver of climate change, such as a change in the concentration of carbon dioxide (CO2) or the output of the sun. We entered the radiative forcing values for each RCP. Line 104-108.

Question: 3. This study includes many abbreviations, such as LUCC, LuccMEBR, IBGE, IMAGE, LuccME, INLAND, LR, and PPAS. The authors can provide a list in the appendix.

Answer: Thanks for the suggestion. Done. Line 331

Question:: 4. There is a reference citation error on line 123 page 6.

Answer: Thanks for the observation. Done. Line 173

Question: 5. In table one, the description of some classes are very detailed, but some are more generalized. Is there a specific reason?.

Answer: Thanks for the observation. The descriptions correspond to the classification and original description of the IBGE land use and land cover classes. Furthermore, some of them have a greater diversity of elements in their compositions. We have inserted the related reference to the description. Table 1

Question: 6. For the study method, the authors used a validation matric (equation 5) to test the similarity between the real and simulated maps. Is this the most common method because the reference paper is 1989. Also, what is the range of this NS index and how to explain it?

Answer: Thanks for the observation. Multiscale analysis is crucial for generating land use and land cover change models as well as their validation. Thus, the metrics adopted for validation should establish the level of similarity between simulated and real maps at different resolutions. For this reason, we adopted the multiscale similarity method adapted from Costanza (1989) and Pontius (2002). It allows both location errors in the resolution of the model itself and spatial pattern errors, degrading the resolution of the maps. We have added a reference to Pontius to the manuscript and rewritten the paragraph to clarify the importance of choosing this metric. Line 229-237. As for the range, the NS value can range from 0\\% (no similarity) to 100\\% (completely similar).

Question: 7. Please increase resolution of figure 3 and 4.

Answer: About the original Fig. 3, we divided it to allow better visualization. Fig. 3 is now represented in Figs. 4 and 5. As for the original Fig. 4, we grouped it; however, we increased the font size and partially restructured it. We chose these formats to avoid an excessive number of figures.

---

## [Decision Letter · Decision Letter 1]

8 Mar 2022

New land-use change scenarios for Brazil: refining global SSPs with a regional spatially-explicit allocation model

PONE-D-21-24083R1

Dear Dr. Gilney Silva Bezerra,

We’re pleased to inform you that your manuscript has been judged scientifically suitable for publication and will be formally accepted for publication once it meets all outstanding technical requirements.

Kind regards,

Eda Ustaoglu, PhD

Academic Editor

PLOS ONE

Additional Editor Comments (optional):

Reviewers' comments:

Reviewer's Responses to Questions

**Comments to the Author**

1. If the authors have adequately addressed your comments raised in a previous round of review and you feel that this manuscript is now acceptable for publication, you may indicate that here to bypass the “Comments to the Author” section, enter your conflict of interest statement in the “Confidential to Editor” section, and submit your "Accept" recommendation.

Reviewer #1: All comments have been addressed

Reviewer #2: All comments have been addressed

2. Is the manuscript technically sound, and do the data support the conclusions?

Reviewer #1: Yes

Reviewer #2: Yes

3. Has the statistical analysis been performed appropriately and rigorously? 

Reviewer #1: Yes

Reviewer #2: Yes

4. Have the authors made all data underlying the findings in their manuscript fully available?

Reviewer #1: Yes

Reviewer #2: Yes

5. Is the manuscript presented in an intelligible fashion and written in standard English?

Reviewer #1: Yes

Reviewer #2: Yes

6. Review Comments to the Author

Reviewer #1: The authors have addressed all me previous comments, providing in this version further details regarding the statistical analysis (Table 5) and details (hypothesis) for the scenario selected. The paper can be accepted, as it will make significant contribution to the research area.

Reviewer #2: The Authors have addressed all of my concerns with the original manuscript. The revised manuscript is ready for publication.

7. PLOS authors have the option to publish the peer review history of their article (what does this mean?). If published, this will include your full peer review and any attached files.

Reviewer #1: No

Reviewer #2: No

---

## [Editor Report · Acceptance letter]

24 Mar 2022

PONE-D-21-24083R1 

New land-use change scenarios for Brazil: refining global SSPs with a regional spatially-explicit allocation model 

Dear Dr. Silva Bezerra:

I'm pleased to inform you that your manuscript has been deemed suitable for publication in PLOS ONE. Congratulations! Your manuscript is now with our production department. 

Kind regards, 

on behalf of

Dr. Eda Ustaoglu 

Academic Editor

PLOS ONE